# Dual Adversarial Semantics-Consistent Network for Generalized Zero-Shot Learning

**Jian Ni[1]**
nj1@mail.ustc.edu.cn

**Shanghang Zhang[2]**
shanghaz@andrew.cmu.edu

**Haiyong Xie[3,4,1]**
haiyong.xie@ieee.org

[1]University of Science and Technology of China, Anhui 230026, China
[2]Carnegie Mellon University, Pittsburgh, PA 15213, USA
[3]Advanced Innovation Center for Human Brain Protection, Capital Medical University, Beijing 100054, China
[4]National Engineering Laboratory for Public Safety Risk Perception and Control by Big Data (NEL-PSRPC), Beijing 100041, China

## Abstract

Generalized zero-shot learning (GZSL) is a challenging class of vision and knowledge transfer problems in which both seen and unseen classes appear during testing. Existing GZSL approaches either suffer from semantic loss and discard discriminative information at the embedding stage, or cannot guarantee the visual-semantic interactions. To address these limitations, we propose a *Dual Adversarial Semantics-Consistent Network* (referred to as *DASCN*), which learns both primal and dual Generative Adversarial Networks (GANs) in a unified framework for GZSL. In DASCN, the primal GAN learns to synthesize inter-class discriminative and semantics-preserving visual features from both the semantic representations of seen/unseen classes and the ones reconstructed by the dual GAN. The dual GAN enforces the synthetic visual features to represent prior semantic knowledge well via semantics-consistent adversarial learning. To the best of our knowledge, this is the first work that employs a novel dual-GAN mechanism for GZSL. Extensive experiments show that our approach achieves significant improvements over the state-of-the-art approaches.

## 1 Introduction

In recent years, tremendous progress has been achieved across a wide range of computer vision and machine learning tasks with the introduction of deep learning. However, conventional deep learning approaches rely on large amounts of labeled data, thus may suffer from performance decay in problems where only limited training data are available. The reasons are two folds. On the one hand, objects in the real world have a long-tailed distribution, and obtaining annotated data is expensive. On the other hand, novel categories of objects arise dynamically in nature, which fundamentally limits the scalability and applicability of supervised learning models for handling this dynamic scenario when labeled examples are not available.

Tackling such restrictions, zero-shot learning (ZSL) has been researched widely recently, recognized as a feasible solution [16, 24]. ZSL is a learning paradigm that tries to fulfill the ability to correctly categorize objects from previous unseen classes without corresponding training samples. However, conventional ZSL models are usually evaluated in a restricted setting where test samples and the search space are limited to the unseen classes only, as shown in Figure 1. To address the shortcomings of ZSL, GZSL has been considered in the literature since it not only learns information that can be transferred to an unseen class but can also generalize to new data from seen classes well.

ZSL approaches typically adopt two commonly used strategies. The first strategy is to convert tasks into visual-semantic embedding problems [4, 23, 26, 33]. They try to learn a mapping function from

the visual space to the semantic space (note that all the classes reside in the semantic space), or to a latent intermediate space, so as to transfer knowledge from the seen classes to the unseen classes. However, the ability of these embedding-based ZSL models to transfer semantic knowledge is limited by the semantic loss and the heterogeneity gap [4]. Meanwhile, since the ZSL model is only trained with the labeled data from the seen classes, it is highly biased towards predicting the seen classes [3]. The second strategy ZSL approaches typically adopt is to use generative methods to generate various visual features conditioned on semantic feature vectors [7, 10, 19, 29, 35], which circumvents the need for labeled samples of unseen classes and boosts the ZSL classification accuracy. Nevertheless, the performance of these methods is limited either by capturing the visual distribution information via only a unidirectional alignment from the class semantics to the visual feature only, or by adopting just a single Euclidean distance as the constraint to preserve the semantic information between the generated high-level visual features and real semantic features. Recent work has shown that the performance of most ZSL approaches drops significantly in the GZSL setting [28].

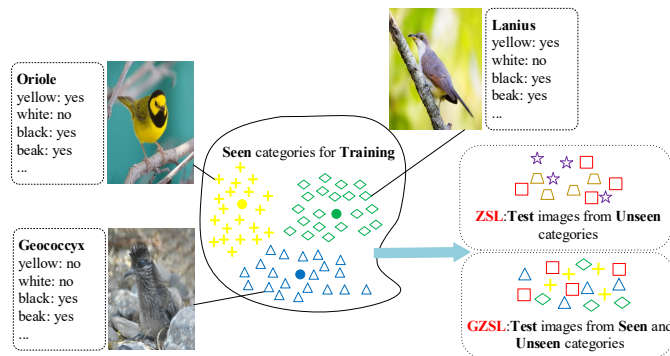

Figure 1: Problem illustration of zero-shot learning (ZSL) and generalized zero-shot learning (GZSL).

To address these limitations, we propose a novel Dual Adversarial Semantics-Consistent Network (referred to as DASCN) for GZSL. DASCN is based on the Generative Adversarial Networks (GANs), and is characterized by its dual structure which enables bidirectional synthesis by allowing both the common visual features generation and the corresponding semantic features reconstruction, as shown in Figure 2. Such bidirectional synthesis procedures available in DASCN boost these two tasks jointly and collaboratively, preserving the visual-semantic consistency. This results in two advantages as follows. First, our generative model synthesizes inter-class discrimination visual features via a classification loss constraint, which makes sure that synthetic visual features are discriminative enough among different classes. Second, our model encourages the synthesis of visual features that represent their semantic features well and are of a highly discriminative semantic nature from the perspectives of both form and content. From the form perspective, the semantic reconstruction error between the synthetic semantic features (reconstructed by the dual GAN from the pseudo visual features generated by the primal GAN) and the real corresponding semantic features is minimized to ensure that the reconstructed semantic features are tightly centered around the real corresponding class semantics. From the content perspective, the pseudo visual features (generated via the primal GAN by further exploiting the reconstructed semantic features as input) are constrained to be as close as possible to their respective real visual features in the data distribution. Therefore, our approach can ensure that the reconstructed semantic features are consistent with the relevant real semantic knowledge, thus avoiding semantic loss to a large extent.

We summarize our contributions as follows. First, we propose a novel generative dual adversarial architecture for GZSL, which preserves semantics-consistency effectively with a bidirectional alignment, alleviating the issue of semantic loss. To the best of our knowledge, DASCN is the first network to employ the dual-GAN mechanism for GZSL. Second, by combining the classification loss and the semantics-consistency adversarial loss, our model generates high-quality visual features with inter-class separability and a highly discriminative semantic nature, which is crucial to the generative approaches used in GZSL. Last but no least, we conduct comprehensive experiments demonstrating that DASCN is highly effective and outperforms the state-of-the-art GZSL methods consistently.

The remainder of this paper is structured as follows. We discuss the related work in Section 2, present our DASCN model in Section 3, evaluate the proposed model in Section 4, and conclude in Section 5.

## 2 Related Work

### 2.1 Zero-Shot Learning

Some of the early ZSL works make use of the primitive attributes prediction and classification, such as DAP [15], and IAP [16]. Recently, the attribute-based classifier has evolved into the embedding-based framework, which now prevails due to its simple and effective paradigm [1, 23, 24, 25, 33]. The core of such approaches is to learn a projection from visual space to semantic space spanned by class attributes [23, 24], or conversely [33], or jointly learn an appropriate compatibility function between the visual space and the semantic space [1, 25].

The main disadvantage of the above methods is that the embedding process suffers from semantic loss and the lack of visual training data for unseen classes, thus biasing the prediction towards the seen classes and undermining seriously the performance of models in the GZSL setting. More recently, generative approaches are promising for GZSL setting by generating labeled samples for the seen and unseen classes. [10] synthesize samples by approximating the class conditional distribution of the unseen classes based on learning that of the seen classes. [29, 35] apply GAN to generate visual features conditioned on class descriptions or attributes, which ignore the semantics-consistency constraint and allow the production of synthetic visual features that may be too far from the actual distribution. [7] consider minimizing L2 norm between real semantics and reconstructed semantics produced by a pre-trained regressor, which is rather weak and unreliable to preserve high-level semantics via the Euclidean distance.

DASCN differs from the above approaches in that it learns the semantics effectively via multi-adversarial learning from both the form and content perspectives. Note that ZSL is also closely related to domain adaptation and image-to-image translation tasks, where all of them assume the transfer between source and target domains. Our approach is motivated by, and is similar in spirit to, recent work on synthesizing samples for GZSL [29] and unpaired image-to-image translation [11, 30, 34]. DASCN preserves the visual-semantic consistency by employing dual GANs to capture the visual and semantic distributions, respectively.

### 2.2 Generative Adversarial Networks

As one of the most promising generative models, GANs have achieved a series of impressive results. The idea behind GANs is to learn a generative model to capture an arbitrary data distribution via a max-min training procedure, which consists of a generator and a discriminator that work against each other. DCGAN [22] extends GAN by leveraging deep convolution neural networks. InfoGAN [5] maximizes the mutual information between the latent variables and generator distribution. In our work, given stabilizing training behavior and eliminating model collapse as much as possible, we utilize WGANs [9] as basic models in a dual structure.

## 3 Methodology

In this section, we first formalize the GZSL task in Section 3.1. Then we present our model and architecture in Section 3.2. We then describe in detail our model's objective, training procedures and generalized zero-shot recognition in Section 3.3, Section 3.4 and Section 3.5, respectively.

### 3.1 Formulation

We denote by $D^{Tr} = \{(x, y, a)|x \in \mathcal{X}, y \in \mathcal{Y}^s, a \in \mathcal{A}\}$ the set of $N^s$ training instances of the seen classes. Note that $x \in \mathcal{X} \subseteq \mathbb{R}^K$ represents $K$-dimensional visual features extracted from convolution neural networks, $\mathcal{Y}^s$ denotes the corresponding class labels, and $a \in \mathcal{A} \subseteq \mathbb{R}^L$ denotes semantic features (*e.g.*, the attributes of seen classes). In addition, we have a disjoint class label set $\mathcal{U} = \{(y, a)|y \in \mathcal{Y}^u, a \in \mathcal{A}\}$ of unseen classes, where visual features are missing. Given $D^{Tr}$ and $\mathcal{U}$, in GZSL, we learn a prediction: $\mathcal{X} \rightarrow \mathcal{Y}^s \cup \mathcal{Y}^u$. Note that our method is of the inductive school where model has access to neither visual nor semantic information of unseen classes in the training phase.

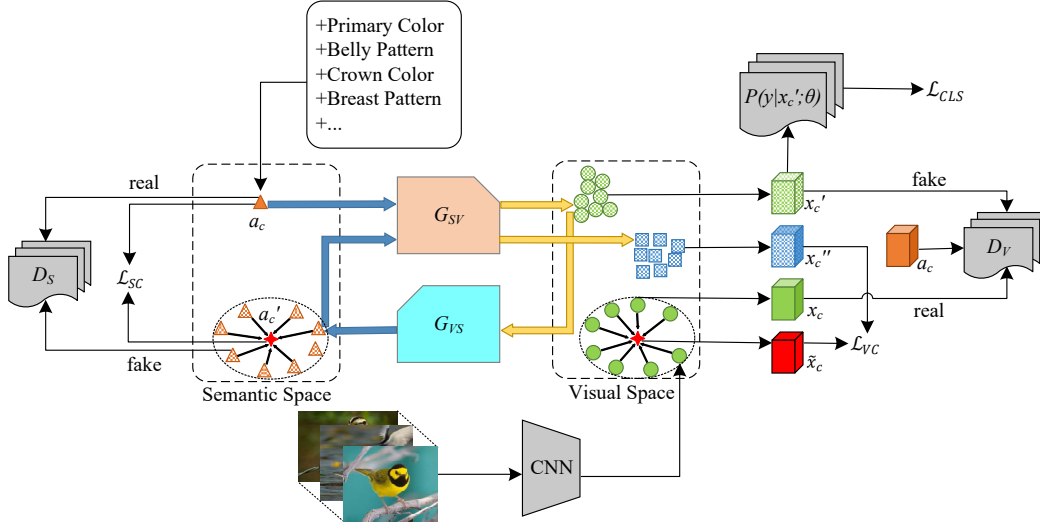

Figure 2: Network architecture of DASCN. The semantic feature of class $c$, represented as $a_c$, and a group of randomly sampled noise vectors are utilized by generator $G_{SV}$ to synthesize pseudo visual features $x_c'$. Then the synthesized visual features are used by generator $G_{VS}$ and discriminator $D_V$ simultaneously to perform semantics-consistency constraint in the perspective of form and content and distinguish between real visual features $x_c$ and synthesized visual features $x_c'$. $D_S$ denotes the discriminator that distinguishes between $a_c$, and the reconstructed semantic feature $a_c'$ generated from corresponding $x_c'$. $x_c''$ are produced by generator $G_{SV}$ taking $a_c'$ and sampled noise as input to perform visual consistency constraint. Please zoom to view better.

## 3.2 Model Architecture

Given the training data $D^{Tr}$ of the seen classes, the primal task of DASCN is to learn a generator $G_{SV}: \mathcal{Z} \times \mathcal{A} \to \mathcal{X}$ that takes the random Gaussian noise $z \in \mathcal{Z}$ and semantic attribute $a \in \mathcal{A}$ as input to generate the visual feature $x' \in \mathcal{X}$, while the dual task is to train an inverse generator $G_{VS}: \mathcal{X} \to \mathcal{A}$. Once the generator $G_{SV}$ learns to generate visual features of the seen classes conditioned on the seen class-level attributes, it can also generate that of the unseen classes. To realize this, we employ two WGANs, the primal GAN and the dual GAN. The primal GAN consists of the generator $G_{SV}$ and the discriminator $D_V$ that discriminates between fake visual features generated by $G_{SV}$ and real visual features. Similarly, the dual GAN learns a generator $G_{VS}$ and a discriminator $D_S$ that distinguishes the fake semantic features generated by $G_{VS}$ from the real data.

The overall architecture and data flow are illustrated in Figure 2. In the primal GAN, we hallucinate pseudo visual features $x_c' = G_{SV}(a_c, z)$ of the class $c$ using $G_{SV}$ based on corresponding class semantic features $a_c$ and then put the real visual features and synthetic features from $G_{SV}$ into $D_V$ to be evaluated. To ensure that $G_{SV}$ generates inter-class discrimination visual features, inspired by that work [29], we design a classifier trained on the real visual features and minimize the classification loss over the generated features. It is formulated as:

$$\mathcal{L}_{CLS} = -E_{x' \sim Px'}[log P(y|x'; \theta)] \tag{1}$$

where $x'$ represents the generated visual feature, $y$ is the class label of $x'$, the conditional probability $P(y|x'; \theta)$ is computed by a linear softmax classifier parameterized by $\theta$.

Following that is one of the main innovations in our work that we guarantee semantics-consistency in both form and content perspectives thanks to dual structure. In form, $G_{SV}(a, z)$ are translated back to semantic space using $G_{VS}$, which outputs $a' = G_{VS}\big(G_{SV}(a, z)\big)$ as the reconstruction of $a$. To achieve the goal that the generated semantic features of each class are distributed around the corresponding true semantic representation, we design the centroid regularization that regularizes the mean of generated semantic features of each class to approach respectively real semantic embeddings

so as to maintain semantics-consistency to a large extent. The regularization is formulated as:

$$\mathcal{L}_{SC} = \frac{1}{C} \sum_{c=1}^{C} \left\| E_{a_{c'} \sim P_{a'}^c}[a_c'] - a_c \right\|_2 \tag{2}$$

where $C$ is the number of seen classes, $a_c$ is the semantic feature of class $c$, $P_{a'}^c$ denotes the conditional distribution of generated semantic features of class $c$, $a_c'$ are the generated features of class $c$ and the centroid is formulated as:

$$E_{a_{c'} \sim P_{a'}^c}[a_c'] = \frac{1}{N_c^s} \sum_{i=1}^{N_c^s} G_{VS}\big(G_{SV}(a_c, z_i)\big) \tag{3}$$

where $N_c^s$ is the number of generated semantic features of class $c$. We employ the centroid regularization to encourage $G_{VS}$ to reconstruct semantic features of each seen class that statistically match real features of that class. From the content point of view, the question of how well the pseudo semantic features $a'$ are reconstructed can be translated into the evaluation of the visual features obtained by $G_{SV}$ taking $a'$ as input. Motivated by the observation that visual features have a higher intra-class similarity and relatively lower inter-class similarity, we introduce the visual consistency constraint:

$$L_{VC} = \frac{1}{C} \sum_{c=1}^{C} \left\| E_{x_{c''} \sim P_{x''}^c}[x_c''] - E_{x_c \sim P_x^c}[x_c] \right\|_2 \tag{4}$$

where $x_c$ is the visual features of class $c$, $x_c''$ is the pseudo visual feature generated by generator $G_{SV}$ employing $G_{VS}\big(G_{SV}(a_c, z)\big)$ as input, $P_x^c$ and $P_{x''}^c$ are conditional distributions of real and synthetic features respectively and the centroid of $x_c''$ is formulated as:

$$E_{x_{c''} \sim P_{x''}^c}[x_c''] = \frac{1}{N_c^s} \sum_{i=1}^{N_c^s} G_{SV}\Big(G_{VS}\big(G_{SV}(a_c, z_i)\big), z_i'\Big) \tag{5}$$

It is worth nothing that our model is constrained in terms of both form and content aspects to achieve the goal of retaining semantics-consistency and achieves superior results in extensive experiments.

### 3.3 Objective

Given the issue that the Jenson-Shannon divergence optimized by the traditional GAN leads to instability during training, our model is based on two WGANs that leverage the Wasserstein distance between two distributions as the objectives. The corresponding loss functions used in the primal GAN are defined as follows. First,

$$L_{D_V} = E_{x' \sim Px'}\big[D_V(x', a)\big] - E_{x \sim P_{data}}\big[D_V(x, a)\big] + \lambda_1 E_{\hat{x} \sim P_{\hat{x}}}\Big[\big(\left\| \nabla_{\hat{x}} D_V(\hat{x}, a) \right\|_2 - 1\big)^2\Big] \tag{6}$$

where $\hat{x} = \alpha x + (1 - \alpha)x'$ with $\alpha \sim U(0,1)$, $\lambda_1$ is the penalty coefficient, the first two terms approximate Wasserstein distance of the distributions of fake features and real features, the third term is the gradient penalty. Second, the loss function of the generator of the primal GAN is formulated as:

$$L_{G_{SV}} = -E_{x' \sim Px'}\big[D_V(x', a)\big] - E_{a' \sim Pa'}\big[D_V(x'', a')\big] + \lambda_2 L_{CLS} + \lambda_3 L_{VC} \tag{7}$$

where the first two terms are Wasserstein loss, the third term is the classification loss corresponding to class labels, the forth term is visual consistency constraint introduced before, and $\lambda_1, \lambda_2, \lambda_3$ are hyper-parameters.

Similarly, the loss functions of the dual GAN are formulated as:

$$L_{D_S} = E_{a' \sim Pa'}\big[D_S(a')\big] - E_{a \sim Pa}\big[D_S(a)\big] + \lambda_4 E_{\hat{y} \sim P\hat{y}}\Big[\big(\left\| \nabla_{\hat{y}} D_S(\hat{y}) \right\|_2 - 1\big)^2\Big] \tag{8}$$

$$L_{G_{VS}} = -E_{a' \sim Pa'}\big[D_S(a')\big] + \lambda_5 L_{SC} + \lambda_6 L_{VC} \tag{9}$$

In Eq. (8) and Eq. (9), $\hat{y} = \beta a + (1 - \beta)a'$ is the linear interpolation of the real semantic feature $a$ and the fake $a'$, and $\lambda_4, \lambda_5, \lambda_6$ are hyper-parameters weighting the constraints.

Table 1: Datasets used in our experiments, and their statistics

| Dataset | Semantics/Dim | # Image | # Seen Classes | # Unseen Classes |
|---------|---------------|---------|----------------|------------------|
| CUB | A/312 | 11788 | 150 | 50 |
| SUN | A/102 | 14340 | 645 | 72 |
| AWA1 | A/85 | 30475 | 40 | 10 |
| aPY | A/64 | 15339 | 20 | 12 |

## 3.4 Training Procedure

We train the discriminators to judge features as real or fake and optimize the generators to fool the discriminator. To optimize the DASCN model, we follow the training procedure proposed in WGAN [9]. The training procedure of our framework is summarized in Algorithm 1. In each iteration, the discriminators $D_V$, $D_S$ are optimized for $n_1$, $n_2$ steps using the loss introduced in Eq. (6) and Eq. (8) respectively, and then one step on generators with Eq. (7) and Eq. (9) after the discriminators have been trained. According to [30], such a procedure enables the discriminators to provide more reliable gradient information. The training for traditional GANs suffers from the issue that the sigmoid cross-entropy is locally saturated as discriminator improves, which may lead to vanishing gradient and need to balance discriminator and generator carefully. Compared to the traditional GANs, the Wasserstein distance is differentiable almost everywhere and demonstrates its capability of extinguishing mode collapse. We put the detailed algorithm for training DASCN model in the supplemental material.

## 3.5 Generalized Zero-Shot Recognition

With the well-trained generative model, we can elegantly generate labeled exemplars of any class by employing the unstructured component $z$ resampled from random Gaussian noise and the class semantic attribute $a_c$ into the $G_{SV}$. An arbitrary number of visual features can be synthesized and those exemplars are finally used to train any off-the-shelf classification model. For simplicity, we adopt a softmax classifier. Finally, the prediction function for an input test visual feature $v$ is:

$$f(v) = arg\max_{y \in \tilde{\mathcal{Y}}} P(y|v; \theta')$$
(10)

where $\tilde{\mathcal{Y}} = \mathcal{Y}^s \cup \mathcal{Y}^u$ for GZSL.

# 4 Experiments

## 4.1 Datasets and Evaluation Metrics

To test the effectiveness of the proposed model for GZSL, we conduct extensive evaluations on four benchmark datasets: CUB [27], SUN [21], AWA1 [15], aPY [6] and compare the results with state-of-the-art approaches. Statistics of the datasets are presented in Table 1. For all datasets, we extract 2048 dimensional visual features via the 101-layered ResNet from the entire images, which is the same as [29]. For fair comparison, we follow the training/validation/testing split as described in [28].

During test time, in the GZSL setting, the search space includes both the seen and unseen classes, i.e. $\mathcal{Y}^u \cup \mathcal{Y}^s$. To evaluate the GZSL performance over all classes, the following measures are applied. (1) ts: average per-class classification accuracy on test images from the unseen classes with the prediction label set being $\mathcal{Y}^u \cup \mathcal{Y}^s$. (2) tr: average per-class classification accuracy on test images from the seen classes with the prediction label set being $\mathcal{Y}^u \cup \mathcal{Y}^s$. (3) H: the harmonic mean of above defined tr and ts, which is formulated as $H = (2 \times ts \times tr)/(ts + tr)$ and quantities the aggregate performance across both seen and unseen test classes. We hope that our model is of high accuracy on both seen and unseen classes.

## 4.2 Implementation Details

Our implementation is based on PyTorch. DASCN consists of two generators and two discriminators: $G_{SV}$, $G_{VS}$, $D_V$, $D_S$. We train specific models with appropriate hyper-parameters. Due to the space

Table 2: Evaluations on four benchmark datasets. *indicates that Cycle-WGAN employs 1024-dim per-class sentences as class semantic rather than 312-dim per-class attributes on CUB, whose results on CUB may not be directly comparable with others.

| Method | AWA1 | | | SUN | | | CUB | | | aPY | | |
|---|---|---|---|---|---|---|---|---|---|---|---|---|
| | ts | tr | H | ts | tr | H | ts | tr | H | ts | tr | H |
| CMT [24] | 0.9 | 87.6 | 1.8 | 8.1 | 21.8 | 11.8 | 7.2 | 49.8 | 12.6 | 1.4 | **85.2** | 2.8 |
| DEVISE [8] | 13.4 | 68.7 | 22.4 | 16.9 | 27.4 | 20.9 | 23.8 | 53.0 | 32.8 | 4.9 | 76.9 | 9.2 |
| ESZSL [23] | 6.6 | 75.6 | 12.1 | 11.0 | 27.9 | 15.8 | 12.6 | 63.8 | 21.0 | 2.4 | 70.1 | 4.6 |
| SJE [1] | 11.3 | 74.6 | 19.6 | 14.7 | 30.5 | 19.8 | 23.5 | 59.2 | 33.6 | 3.7 | 55.7 | 6.9 |
| SAE [13] | 1.8 | 77.1 | 3.5 | 8.8 | 18.0 | 11.8 | 7.8 | 54.0 | 13.6 | 0.4 | 80.9 | 0.9 |
| LESAE [18] | 19.1 | 70.2 | 30.0 | 21.9 | 34.7 | 26.9 | 24.3 | 53.0 | 33.3 | 12.7 | 56.1 | 20.1 |
| SP-AEN [4] | - | - | - | 24.9 | 38.2 | 30.3 | 34.7 | **70.6** | 46.6 | 13.7 | 63.4 | 22.6 |
| RN [25] | 31.4 | **91.3** | 46.7 | - | - | - | 38.1 | 61.1 | 47.0 | - | - | - |
| TRIPLE [31] | 27 | 67.9 | 38.6 | 22.2 | 38.3 | 28.1 | 26.5 | 62.3 | 37.2 | - | - | - |
| f-CLSWGAN [29] | 57.9 | 61.4 | 59.6 | 42.6 | 36.6 | 39.4 | 43.7 | 57.7 | 49.7 | - | - | - |
| KERNEL [32] | 18.3 | 79.3 | 29.8 | 19.8 | 29.1 | 23.6 | 19.9 | 52.5 | 28.9 | 11.9 | 76.3 | 20.5 |
| PSR [2] | - | - | - | 20.8 | 37.2 | 26.7 | 24.6 | 54.3 | 33.9 | 13.5 | 51.4 | 21.4 |
| DCN [17] | 25.5 | 84.2 | 39.1 | 25.5 | 37 | 30.2 | 28.4 | 60.7 | 38.7 | 14.2 | 75.0 | 23.9 |
| SE-GZSL [14] | 56.3 | 67.8 | 61.5 | 40.9 | 30.5 | 34.9 | 41.5 | 53.3 | 46.7 | - | - | - |
| GAZSL [35] | 25.7 | 82.0 | 39.2 | 21.7 | 34.5 | 26.7 | 23.9 | 60.6 | 34.3 | 14.2 | 78.6 | 24.1 |
| DASCN (Ours) | **59.3** | 68.0 | **63.4** | 42.4 | **38.5** | **40.3** | **45.9** | 59.0 | **51.6** | **39.7** | 59.5 | **47.6** |
| Cycle-WGAN* [7] | 56.4 | 63.5 | 59.7 | **48.3** | 33.1 | 39.2 | 46.0 | 60.3 | 52.2 | - | - | - |

limitation, here we take CUB as an example. Both the generators and discriminators are MLP with LeakyReLU activation. In the primal GAN, $G_{SV}$ has a single hidden layer containing 4096 nodes and an output layer that has a ReLU activation with 2048 nodes, while the discriminator $D_V$ contains a single hidden layer with 4096 nodes and an output layer without activation. $G_{VS}$ and $D_S$ in the dual GAN have similar architecture with $G_{SV}$ and $D_V$ respectively. We use $\lambda_1 = \lambda_4 = 10$ as suggested in [9]. For loss term contributions, we cross-validate and set $\lambda_2 = \lambda_3 = \lambda_6 = 0.01$, $\lambda_5 = 0.1$. We choose noise $z$ with the same dimensionality as the class embedding. Our model is optimized by Adam [12] with a base learning rate of $1e^{-4}$.

## 4.3 Compared Methods and Experimental Results

We compare DASCN with state-of-the-art GZSL models. These approaches fall into two categories. (1) Embedding-based approaches: CMT [24], DEVISE [8], ESZSL [23], SJE [1], SAE [13], LESAE [18], SP-AEN [4], RN [25], KERNEL [32], PSR [2], DCN [17], TRIPLE [31]. This category suffers from the issue of the bias towards seen classes due to the lack of instances of the unseen classes. (2) Generative approaches: SE-GZSL [14], GAZSL [35], f-CLSWGAN [29], Cycle-WGAN [7]. This category synthesizes visual features of the seen and unseen classes and perform better for GZSL compared to the embedding-based methods.

Table 2 summarizes the performance of all the comparing methods under three evaluation metrics on the four benchmark datasets, which demonstrates that for all datasets our DASCN model significantly improves the ts measure and H measure over the state-of-the-arts. Note that Cycle-WGAN [7] employs per-class sentences as class semantic features on CUB dataset rather than per-class attributes that are commonly used by other comparison methods, so its results on CUB may not be directly comparable with others. On CUB, DASCN achieves 45.9% in ts and 51.6% in H, with improvements over the state-of-the-art 2.2% and 1.9% respectively. On SUN, it obtains 42.4% in ts measure and 40.3% in H measure. On AWA1, our model outperforms the runner-up by a considerable gap in H measure: 1.9%. On aPY, DASCN significantly achieves improvements over the other best competitors 25.5% in ts measure and 23.5% in H measure, which is very impressive. The performance boost is attributed to the effectiveness of DASCN that imitate discriminative visual features of the unseen classes. In conclusion, our model DASCN achieves a great balance between seen and unseen classes classification and consistently outperforms the current state-of-the-art methods for GZSL.

Table 3: Comparison between the reported results of Cycle-WGAN and our model. * indicates employing the same semantic features (per-class sentences (stc)) as Cycle-WGAN on CUB.

|  | FLO | | | CUB* | | | SUN | | | AWA1 | | |
|---|---|---|---|---|---|---|---|---|---|---|---|---|
| Method | ts | tr | H | ts | tr | H | ts | tr | H | ts | tr | H |
| Cycle-WGAN | 59.1 | 71.1 | 64.5 | 46.0 | 60.3 | 52.2 | **48.3** | 33.1 | 39.2 | 56.4 | 63.5 | 59.7 |
| DASCN | **60.5** | **80.4** | **69.0** | **47.4** | 60.1 | **53.0** | 42.4 | **38.5** | **40.3** | **59.3** | 68.0 | **63.4** |

To further clarify the advantages of DASCN over Cycle-WGAN [7] in both methodology and empirical results, we conduct the following experiments: (1) we use the same semantic features (per-class sentences (stc)) as Cycle-WGAN uses for DASCN on the CUB dataset, (2) we add the FLO [6] dataset employed by Cycle-WGAN as a benchmark. As shown in Table 3, results on four benchmarks consistently demonstrate the superiority of DASCN. The main novelty of our work is the integration of dual structure mechanism and visual-semantic consistencies into GAN for bidirectional alignment and alleviating semantic loss. In contrast, Cycle-WGAN only consists of one GAN and a pre-trained regressor, which only minimizes L2 norm between the reconstructed and real semantics. As a result, Cycle-WGAN is rather weak and unreliable to preserve high-level semantics via the Euclidean distance. Compared to that, thanks to the dual-GAN structure and visual-semantic consistencies loss, DASCN explicitly supervises that the generated features have highly discriminative semantic nature on the high-level aspects and effectively preserve semantics via multi-adversarial learning in both form and content perspectives.

More specifically, we build two GANs for visual and semantic generation, and design two consistency regularizations accordingly: (1) semantic consistency to align the centroid of the synthetic semantics and real semantic, (2) visual consistency for not only matching the real visual features but also enforcing synthetic semantics to have highly discriminative nature to further generate effective visual features. Compared to the Cycle-WGAN that only minimizes L2 norm of reconstructed and real semantics, the novelty being introduced is the tailor-made semantic high-level consistency at a finer granularity.

Note that we not only generate synthetic semantic features from the synthetic visual features, but also further generate synthetic visual features again based on the synthetic semantic features, which is constrained by visual consistency loss to ensure the generated features have highly discriminative semantic nature. Such bidirectional synthesis procedures boost the quality of synthesized instances collaboratively via dual structure.

## 4.4 Ablation Study

We now conduct the ablation study to evaluate the effects of the dual structure, the semantic centroid regularization $\mathcal{L}_{SC}$, and the visual consistency constraint $\mathcal{L}_{VC}$. We take the single WGAN model f-CLSWGAN as baseline, and train three variants of our model by keeping the single dual structure or that adding the only semantic or visual constraint, denoted as Dual-WGAN, Dual-WGAN $+\mathcal{L}_{SC}$, Dual-WGAN $+\mathcal{L}_{VC}$, respectively. Table 4 shows the performance of each setting, the performance of the single Dual-WGAN on the H metric drops drastically by 4.9% on aPY, 1.4% on AWA1, 1.3% on CUB and 0.7% on SUN, respectively. This clearly highlights the importance of designed semantic and visual constraints to provide an explicit supervision to our model. In the case of lacking semantic or visual unidirectional constrains, on aPY, our model drops by 1.3% and 3.6% respectively, while on AWA1 the gap are 0.9% and 0.7%. In general, the three variants of our proposed model tend to offer more superior and balanced performance than the baseline. DASCN incorporates dual structure, semantic centroid regularization and visual consistency constraint into a unified framework and achieves the best improvement, which demonstrates that different components promote each other and work together to improve the performance of DASCN significantly.

## 4.5 Quality of Synthesized Samples

We perform an experiment to gain a further insight into the quality of the generated samples, which is one key issue of our approach, although the quantitative results reported for GZSL above demonstrate that the samples synthesized by our model are of significant effectiveness for GZSL task. Specifically, we randomly sample three unseen categories from aPY and visualize both true visual features and

Table 4: Effects of different components on four benchmark datasets with GZSL setting.

| Methods | aPY | | | AWA1 | | | CUB | | | SUN | | |
|---|---|---|---|---|---|---|---|---|---|---|---|---|
| | ts | tr | H | ts | tr | H | ts | tr | H | ts | tr | H |
| WGAN-baseline | 32.4 | 57.5 | 41.4 | 57.9 | 61.4 | 59.6 | 43.7 | 57.7 | 49.7 | 42.6 | 36.6 | 39.4 |
| Dual-WGAN | 34.1 | 57.0 | 42.7 | 57.5 | 67.4 | 62.0 | 44.5 | 57.9 | 50.3 | 42.7 | 36.9 | 39.6 |
| Dual-WGAN $+\mathcal{L}_{SC}$ | 35.4 | 58.2 | 44.0 | 57.7 | **68.6** | 62.7 | 44.9 | 58.5 | 50.8 | 42.9 | 37.3 | 39.9 |
| Dual-WGAN $+\mathcal{L}_{VC}$ | 36.7 | **62.0** | 46.3 | 58.3 | 67.3 | 62.5 | 45.2 | **59.1** | 51.2 | **43.5** | 36.5 | 39.7 |
| DASCN | **39.7** | 59.5 | **47.6** | **59.3** | 68.0 | **63.4** | **45.9** | 59.0 | **51.6** | 42.4 | **38.5** | **40.3** |

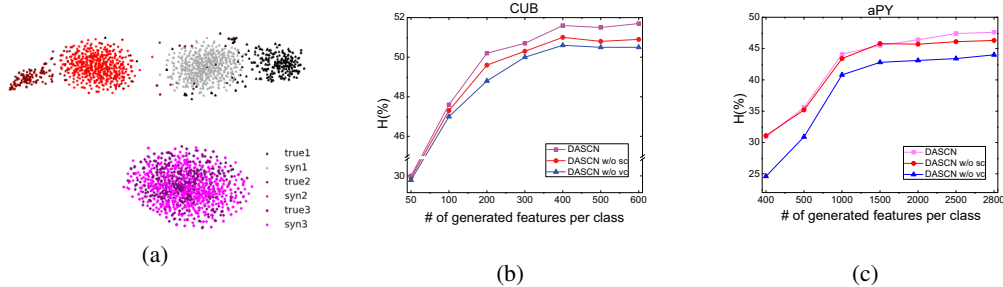

(a)                                    (b)                                    (c)

Figure 3: **(a)**: t-SNE visualization of real visual feature distribution and synthesized feature distribution from randomly selected three unseen classes; **(b, c)**: Increasing the number of samples generated by DASCN and its variants wrt harmonic mean H. DASCN w/o SC denotes DASCN without semantic consistency constraint and DASCN w/o VC stands for that without visual consistency constraint.

synthesized visual features using t-SNE [20]. Figure 3(a) depicts the empirical distributions of the true visual features and the synthesized visual features. We observe the clear patterns of intra-class diversity and inter-class separability in the figure. This intuitively demonstrates that not only the synthesized feature distributions well approximate the true distributions but also our model introduces a high discriminative power of the synthesized features to a large extent.

Finally, we evaluate how the number of the generated samples per class affects the performance of DASCN and its variants. Obviously, as shown in Figure 3(b) and Figure 3(c), we notice not only that H increases with an increasing number of synthesized samples and asymptotes gently, but also that DASCN with visual-semantic interactions achieves better performance in all circumstance, which further validates the superiority and rationality of different components of our model.

## 5 Conclusion

We propose DASCN, a novel generative model for GZSL, to address the challenging problem where existing GZSL approaches either suffer from the semantic loss or cannot guarantee the visual-semantic interactions. DASCN can synthesize inter-class discrimination and semantics-preserving visual features for both seen and unseen classes. The DASCN architecture is novel in that it consists of a primal GAN and a dual GAN to collaboratively promote each other, which captures the underlying data structures of both visual and semantic representations. Thus, our model can effectively enhance the knowledge transfer from the seen categories to the unseen ones, and can effectively alleviate the inherent semantic loss problem for GZSL. We conduct extensive experiments on four benchmark datasets and compare our model against the state-of-the-art models. The evaluation results consistently demonstrate the superiority of DASCN to state-of-the-art GZSL models.

**Acknowledgments**

This research is supported in part by the National Key Research and Development Project (Grant No. 2017YFC0820503), the National Science and Technology Major Project for IND (investigational new drug) (Project No. 2018ZX09201014), and the CETC Joint Advanced Research Foundation (Grant No. 6141B08080101,6141B08010102).

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
