[Supplementary Material · Supplemental_Material.pdf]

# Supplemental Material

## 1 ALGORITHMS FOR TRAINING DASCN

We describe the detailed algorithm for training DASCN as following:

---

**Algorithm 1** Training processing of the proposed DASCN

---

**Input**: the maximal loops $M$, the batch size $m$, the primal GAN with generator parameters $\theta_A$, discriminator parameters $\omega_A$ and iteration $n_1$, the dual GAN with generator parameters $\theta_B$ and discriminator parameters $\omega_B$ and iteration $n_2$.

randomly initialize $\omega_t, \theta_t, t \in A, B$

1: **for** iter=1 to $M$ **do**
2:    **for** i=1 to $n_1$ **do**
3:        Sample a minibatch of $x$, matched $a$, random noise $z$
4:        $x' \leftarrow G_{SV}(a, z)$
5:        Update $\omega_A$ to minimize Eq. (6)
6:    **end for**
7:    **for** j=1 to $n_2$ **do**
8:        Sample random noise $z$
9:        $a' \leftarrow G_{VS}\big(G_{SV}(a, z)\big)$
10:       Update $\omega_B$ to minimize Eq. (8)
11:   **end for**
12:   Sample random noise $z$
13:       $a' \leftarrow G_{VS}\big(G_{SV}(a, z)\big)$
14:   Update $\theta_B$ to minimize Eq. (9)
15:   Sample random noise $z, z'$
16:       $x' \leftarrow G_{SV}(a, z)$
17:       $x'' \leftarrow G_{SV}\Big(G_{VS}\big(G_{SV}(a, z)\big), z'\Big)$
18:   Update $\theta_A$ to minimize Eq. (7)
19: **end for**

---