[Reviews · NeurIPS 2019]

Reviewer 1



(+) The primary contribution of the paper is the dual-GAN structure with semantics-consistency and visual-consistency loss. The paper has novel components (although it is close to [7], see below). The paper shows that their model performs better than the existing methods on GZSL benchmark datasets. (+) The paper is written well and easy to follow. (+/-) There are multiple losses in the paper that contribute to the overall performance. To better understand the individual contribution of these losses, the paper gives an ablation study in Table 3. However, it should also include results of ablation study on CUB and SUN datasets in the main paper. It is important since these two datasets are considerably larger than the other datasets in terms of classes. (-) In Figure 3(b), it appears that DASCN w/o SC has better performance than the full DASCN model. This needs to be clarified. (-) The paper is closely related to [7] but the paper does not have sufficient discussion and comparisons. The main idea of both, this paper and [7], is inspired from the cycle-consistency in image-to-image translation works [33]. In [7] (i) a GAN and a regressor is used instead of the proposed dual-GAN structure and, (ii) the proposed visual-consistency loss (VC) is not used in [7]. To establish the importance of dual-GAN structure, the performance of DASCN w/o VC should be compared with that of [7]. In the present state of paper, this cannot be done because (i) [7] have used semantic features different from others on CUB dataset (ii) authors have not presented the ablation study results on SUN and the CUB dataset. This is a very important issue and should be carefully addressed in the rebuttal. Mistakes and typos: (Important) In the second term of equation (7), it should be Dv(x’’, a’) instead of Dv(x, a’). It should be lowercase c in summation limits in equation (2) Line 157: instability during training Post rebuttal edit: The rebuttal answers some questions, but adds significant results including results on a new dataset which was absent in the original paper (FLO). In comparison with [7] the results on CUB and SUN (the bigger datasets) do not show significant improvements. AWA1 is smaller and FLO has been added afresh. IMO the paper needs to be revised and resubmitted with all the details added and discussions extended/revised.

Reviewer 2



This paper proposes a novel generative model for GZSL to synthesize inter-class discrimination and semantics preserving visual features for seen and unseen classes. The proposed DASCN model preserves the visual-semantic consistency by employing dual GANs to capture the visual and semantic distributions, respectively. Extensive experimental results consistently demonstrates the superiority of DASCN to state-of-the-art GZSL approaches.

Reviewer 3



Originality: low The proposed method looks the variant of Cycle-WAGN [7] that changes the consistency loss to class-wise loss. Quality: middle The proposed method shows better performance by a large margin and ablation study shows that proposed loss works well. One question is the result that apply Cycle-WGAN [7] in the proposed setting because the proposed method is similar to [7]. Another question is that the high performance of the proposed method arises from ts that is the accuracy for unseen class. Therefore I think the contribution is for zero-shot learning part more than generalized zero-shot learning part. If it is true, I am curious about the comparison of the methods in the standard zero-shot learning setting with the state-of-the-art zero-shot learning methods. Clarity: high The paper is easy to follow and the proposed algorithm is well described. Significance: middle Though the proposed method experimentally works well, it is not clear if the performance gain comes from the success of generalized zero-shot learning or simply the success of zero-shot learning. Also, given Cycle-WGAN, the technical contribution seems small. After Rebuttal I increase the score because the rebuttal address my concerns about the difference to Cycle-WGAN and the implication of the score ts.

[Author Response · NeurIPS 2019]

We appreciate the feedback from R1, R2, and R3. We address the questions below and will revise our paper accordingly.
[R1 & R3 Sufficient discussion of the difference and direct empirical comparisons with Cycle-wgan [7]]
The main novelty of our work is the integration of DUAL structure mechanism and visual-semantic consistencies (VC)
into GAN for bidirectional alignment and alleviating semantic loss. In contrast, Cycle-wgan only consists of one GAN
and a pre-trained regressor, which only minimizes L2 norm between the reconstructed and real semantics. Cycle-wgan
is rather weak and unreliable to preserve high-level semantics via the Euclidean distance (Line 88-91). Compared to
that, thanks to the dual-GAN structure and VC loss, DASCN explicitly supervises that the generated features have
highly discriminative semantic nature on the high-level aspects and effectively preserve semantics via multi-adversarial
learning in both form and content (Line 45-62). Specifically, we build two GANs for visual & semantic generation and
two consistency regularization are accordingly devised: 1) semantic consistency to align the centroid of the synthetic
semantics and real semantic, 2) visual consistency for not only matching the real visual features but also enforcing
synthetic semantics to have highly discriminative nature to further generate effective visual features. Compared to
the Cycle-wgan that only minimizes L2 norm of reconstructed & real semantics, the novelty being introduced is the
tailor-made semantic high-level consistency at a finer granularity. We not only generate synthetic semantic features from
the synthetic visual features, but also further generate synthetic visual features again based on the synthetic semantic
features, which is constrained by VC to ensure the generated features have highly discriminative semantic nature. Such
bidirectional synthesis procedures boost the quality of synthesized instances collaboratively via DUAL structure.
To address your comment on direct empirical comparisons, we conducted the following experiments: (1) Compare
Cycle-wgan with DASCN w/o VC, and the full DASCN on four benchmarks. (2) Use the same semantic features
(per-class sentences (stc)) as Cycle-wgan for DASCN on CUB dataset. (3) Add FLO as a benchmark. As shown in Table
1, results on four benchmarks consistently demonstrate the superiority of DASCN. DASCN w/o VC also outperforms
Cycle-wgan in most cases. We will add the discussions and more empirical comparisons in the final version.
Table 1: Comparison between the reported results of Cycle-wgan [7] and our model. * indicates employing the same semantic features (per-class sentences (stc)) as Cycle-wgan on CUB.

|  | FLO | | | CUB* | | | SUN | | | AWA1 | | |
|---|---|---|---|---|---|---|---|---|---|---|---|---|
| Method | ts | tr | H | ts | tr | H | ts | tr | H | ts | tr | H |
| Cycle-wgan [1] | 59.1 | 71.1 | 64.5 | 46.0 | 60.3 | 52.2 | **48.3** | 33.1 | 39.2 | 56.4 | 63.5 | 59.7 |
| DASCN w/o VC | 58.5 | 78.8 | 67.2 | 46.3 | **60.5** | 52.5 | 42.9 | 37.3 | 39.9 | 57.7 | **68.6** | 62.7 |
| DASCN | **60.5** | **80.4** | **69.0** | **47.4** | 60.1 | **53.0** | 42.4 | **38.5** | **40.3** | **59.3** | 68.0 | **63.4** |

[R1 Ablation study on CUB and SUN] We
conducted ablative experiments on CUB and
SUN (Table 2), which demonstrate different
components promote each other and work
together to improve performance of DASCN.
We will add the results in the final version.
[R1 Typo in Figure 3(b) and other mistakes]
We would like to clarify that DASCN model
does perform better than DASCN w/o SC.
There is a typo in the legend of Figure
3(b). The magenta polyline should repre-
sent DASCN while the red one should be DASCN w/o SC. We will correct these typos in the final version.

Table 2: Ablation study on SUN and CUB datasets with GZSL setting.

|  | SUN | | | CUB | | |
|---|---|---|---|---|---|---|
| Methods | ts | tr | H | ts | tr | H |
| WGAN-baseline | 42.6 | 36.6 | 39.4 | 43.7 | 57.7 | 49.7 |
| Dual-WGAN $+\mathcal{L}_{SC}$ | 42.9 | 37.3 | 39.9 | 44.9 | 58.5 | 50.8 |
| Dual-WGAN $+\mathcal{L}_{VC}$ | **43.5** | 36.5 | 39.7 | 45.2 | **59.1** | 51.2 |
| DASCN | 42.4 | **38.5** | **40.3** | **45.9** | 59.0 | **51.6** |

[R3 The contribution is for ZSL part more than GZSL part]
We need to clarify that GZSL is totally different problem from ZSL, and is much more challenging [27]. There is no
inclusive relationship between GZSL and ZSL. ts is measured in GZSL setting, and is not related to the performance
of ZSL. Thus the good performance on ts cannot lead to the conclusion that "The contribution is for ZSL part more
than GZSL part." On the contrary, it exactly indicates the efficacy of DASCN under the GZSL setting. [27] has shown
that performance of existing ZSL methods drops significantly in GZSL setting, for the seen classes are included in the
search space and act as distractors for the instances from unseen classes. DASCN is particularly designed for GZSL to
overcome the shortcomings of existing ZSL methods, which are often biased towards seen classes and undermine ts.
[R3 Results of Cycle-wgan [7] in the proposed setting]
We have provided the direct empirical comparisons with Cycle-wgan in the proposed setting in Section 4.3 (Table 2 in
our paper). On AWA1 and SUN datasets, DASCN has a significant edge over Cycle-wgan. To further address your
comment, we also conducted more experiments to compare DASCN and Cycle-wgan, please refer to the first rebuttal
bullet. Results (in Table 1 above) on four benchmark datasets consistently demonstrate the superiority of DASCN.
[R3 Clarification of the difference to Cycle-wgan [7]]
The difference of DASCN and Cycle-wgan is not "changes the consistency loss to class-wise loss". Please refer to the
first rebuttal bullet, where we discussed the advantages of DASCN over Cycle-wgan in both methodology and empirical
results. We hope to address your questions and sincerely appreciate it a lot if you could update your score accordingly.

[Meta-Review · NeurIPS 2019]

The paper received all accept recommendations and the AC agrees with the recommendation. The authors are requested to revise the paper with the additional clarifications wrt to [7], results from the new experiments on FLO, and discussion points.